# Abdominal Wall Movements Predict Intra-Abdominal Pressure Changes in Rats: A Novel Non-Invasive Intra-Abdominal Pressure Detection Method

**DOI:** 10.3390/children10081422

**Published:** 2023-08-21

**Authors:** Deirdre Vincent, Stefan Mietzsch, Wolfgang Braun, Magdalena Trochimiuk, Konrad Reinshagen, Michael Boettcher

**Affiliations:** 1Department of Pediatric Surgery, University Medical Center Hamburg-Eppendorf, 20246 Hamburg, Germany; 2Fritz Stephan GmbH, 56412 Gackenbach, Germany; 3Department of Pediatric Surgery, University Medical Center Mannheim, Heidelberg University, 69117 Heidelberg, Germany

**Keywords:** acute abdomen, animal model, Graseby capsule, intra-abdominal pressure, neonatal intensive care

## Abstract

(1) Background: As increases in intra-abdominal pressure (IAP) result in irreversible tissue damage, monitoring IAP in critically ill patients using the common urinary bladder catheter method is essential. However, this method can result in complications and is not suitable for very low birth weight neonates. The aim of this study was to establish a non-invasive and accurate method to detect IAP changes using an animal model. (2) Methods: IAP changes via intra-abdominal air application (up to 20 mmHg) were measured in 19 Wistar rats via an intra-abdominally placed intracranial pressure probe. Concurrently, abdominal surface tension was measured using a Graseby capsule (GC). (3) Results: A high correlation between abdominal wall distension and IAP (r = 0.9264, CI 0.9249–0.9279) was found for all subjects. (4) Conclusions: IAP changes in rats can be detected non-invasively using a GC. However, further studies are necessary to assess whether IAP changes can be measured using a GC in the neonatal population.

## 1. Introduction

Because increases in intra-abdominal pressure (IAP) may result in elevated morbidity and mortality rates, the assessment and monitoring of IAP in critically ill patients is standard practice throughout intensive care units (ICUs) [1]. This is due to the abdominal compartment’s limited compliance; where IAP increases beyond 20 mmHg can lead to acute events, such as abdominal compartment syndrome (ACS), which often results in multisystem organ failure [2]. Consequently, ACS is associated with high morbidity and mortality rates, particularly within the pediatric and neonatal populations [3].

The causes of IAP increases can be divided into (1) tissue edema or (2) intraperitoneal fluid collection. Thus, a common trigger for IAP elevation is abdominal trauma and the subsequent blood or fluid collection or edema formation. However, non-traumatic events can also result in IAP increases; bowel inflammation, ischemia, and infarction are associated with the swelling of cells and fluid collection, which can progress to edema formation. Once the abdominal compartment pressure overcomes the pressure inside the capillaries, the perfusion of organs decreases, resulting in ischemia and infarction [4]. 

One of the main challenges associated with diagnosing ACS is that a high level of clinical expertise is typically necessary, as symptoms are unspecific [5]. Further, the confirmation of suspicion of ACS generally requires measurements of above-average IAP values, which are usually obtained via a urinary bladder pressure measurement. Normal IAP ranges from 0 to 5 mmHg, with physiological compromises occurring when IAP rises above 8–10 mmHg. At this stage, symptoms may not be present, but the progression to IAP values of >20 mmHg might happen quickly [6]. Thus, early recognition of rising IAP is critical, as prompt intervention and treatment can help prevent ACS from developing.

As ACS is associated with high morbidity and mortality rates, monitoring IAP in ICUs, particularly in pediatric and neonatal ICUs (NICUs), should be considered as part of standard procedures, as it is indicative of problems arising after reduction surgery for abdominal wall defects or corrective surgery for congenital diaphragmatic hernias, volvulus, and necrotizing enterocolitis (NEC). In fact, NEC, one of the most common and severe gastrointestinal diseases, predominantly afflicting premature neonates, has been linked to ACS [7]. 

Currently, the gold standard for measuring IAP in both adults and children is via a self-retaining urinary balloon catheter, also known as a Foley-type catheter [8]. Although this method is well established and inexpensive, the procedure is invasive and associated with non-negligible risks. Furthermore, in very low or extremely low birth weight neonates, the diameter of even the smallest Foley catheter can be too large to place without harming the neonate and, consequently, IAP measurements cannot be obtained [9]. This dilemma particularly applies to neonates at risk of developing NEC, as the incidence rates for NEC increase with prematurity and decreasing birth weight [10]. As such, there is a need for a new and preferably non-invasive IAP monitoring method for neonates.

A device commonly used to aid NICU patients’ ventilation settings, is the so-called Graseby capsule (GC), which provides a reliable signal of spontaneous respiratory activity, that is only minimally influenced by ventilation [11]. The GC, a foam-filled flexible capsule, is applied to the abdominal wall subxiphisternally and measures small changes in abdominal wall distension. Due to the GC’s ability to detect even small changes in distension, the aim of this study was to evaluate the capsule’s capability to detect IAP changes. Therefore, a rat animal testing model, representing preterm neonates, was developed to compare IAP measurements using an intra-abdominally placed intracranial pressure probe (ICP) in combination with the GC. We hypothesized that continuous monitoring of abdominal wall distension using the GC can be used to non-invasively detect changes in IAP.

## 2. Materials and Methods

### 2.1. Study Design

A total of 19 Wistar rats were utilized in this animal model. This study was approved by the Hamburg State Administration for animal research (72/14). All environmental parameters within the animal facility complied with the German guide for care and use of laboratory animals (Tierschutzgesetz).

### 2.2. Animal Procedures

#### 2.2.1. Animals

Wistar rats were received on day 12 postpartum and housed within the animal care facility (temperature 20–24 °C, relative humidity 50–60%, controlled light conditions—12 h of light, food and water ad libitum). 

#### 2.2.2. Equipment and Instruments

The following materials were used in the experiment:Graseby capsule (GC);Codman intracranial pressure (ICP) probe;Heidelberger extension, syringe, and clamp;Tissue adhesive (LIQUIBAND Standard, Advanced Medical Solutions Ltd., Plymouth, UK) and suturing material;Anesthesia and surgical equipment.

#### 2.2.3. Procedure

Prior to surgery, all animals were anesthetized via the inhalation of 5% isoflurane gas solution in a chamber. Sedation depth was evaluated via the assessment of the rat’s reflex to painful stimuli. If necessary, a second dose of the anesthetic was administered to ensure adequate anesthesia and sedation depth. Anesthesia was maintained using a 2.5% solution of isoflurane gas administered via a nose mask. The rat was then placed in a dorsal decubitus position, with all four limbs fixed to the surgical table using simple rubber bands. Preoperative depilation of the thorax and abdomen was performed (hair trimmer), followed by antisepsis using polyvinylpyrrolidone-iodine. 

A small laparotomy (<1 cm) was made on either side of the lower abdomen, after which we placed the Heidelberger extension in the left (Figure 1B), and the Codman ICP probe in the right lower quadrant (Figure 1A). Fixation of the Heidelberger extension and Codman ICP probe was performed as follows: prior to fixation, the extension and probe were tunneled, and fascial closure via suturing was conducted to prevent displacement and leakage. Cutaneous closure was conducted using single sutures and tissue adhesive. The GC was then placed onto the abdomen and held in place using surgical tape (Figure 1C). 

The Codman ICP probe was connected to its standard monitoring device, and the GC was joined to an amplifier (National instruments). Values measured using the Codman ICP probe were measured in mmHg. To assess changes in IAP and abdominal movement/surface tension, pressure was applied via the Heidelberger extension using a syringe filled with air in increments of (1) 1 mmHg until 10 mmHg and (2) 2 mmHg until 20 mmHg for a period of 2 min. Changes in abdominal movement/surface tension, as well as IAP values, were continuously monitored throughout the experiment in an automatic fashion. The experiment was terminated once a value of 20 mmHg was measured using the Codman ICP probe or if a subject showed any signs of distress. Euthanasia with isoflurane via cervical dislocation was performed after the experiment was terminated and while the animal was still sedated.

### 2.3. Statistics

All data were analyzed using SPSS Statistics 28 (IBM). As this was a pilot study, no power calculation was performed. For the correlation, Pearson’s R was calculated. Baseline characteristics were analyzed using a one-sample *t* test. The level of significance was set at <0.05.

## 3. Results

In total, 19 female rats were included in the study. They exhibited similar characteristics, like age (29.05 (0.94) weeks) and weight (412.40 (43.91) g). As shown in Figure 2, the results suggest a very strong positive correlation between changes in abdominal wall distension and IAP (r = 0.9264, CI 0.9249–0.9279). This means that, as IAP was amplified, a proportional increase in readings obtained using the GC was observed. 

## 4. Discussion

The current study confirms that GCs can detect changes in IAP in rats. Not only was there a positive correlation between increases in IAP and increases in GC readings, but increasing the IAP through the application of air to the abdomen also caused a proportional rise in GC measurements. As such, we can assume that values obtained using a GC accurately measure small changes in IAP. 

The GC was developed in the late 1970s in England by Wright and Callan [12] in response to increased rates of “sudden infant death” among babies and is now commonly used for the detection of central apnea. The GC makes use of pneumatic signals that are created by the deformation of a flexible and foam-filled capsule fixed, using tape, to the abdominal wall, where it is applied to the subxiphisternum. 

Throughout their use in NICUs worldwide, GCs have been shown to provide a reliable signal of spontaneous respiratory activity and are minimally influenced by a ventilator. The capsule makes use of premature infants’ highly compliant chest walls and poorly compliant lungs. Therefore, diaphragmatic contraction (clinically observed as recession) results in xiphisternal retractions in premature neonates. These contractions, in turn, distort the capsules. However, ventilator inflation leaves the capsules undistorted [13]. Thus, what makes GCs of interest for measuring IAP changes, is not only that they are non-invasive, but also that the devices are already used in NICUs to aid in respiration and ventilation settings [11], making the capsule’s alternate use for IAP measurement easily implementable. Furthermore, patients are not harmed through the application of GCs and changes in IAP can be measured continuously without further manipulation of the patient. 

This is particularly important as continuous monitoring of IAP, when performed using the current gold standard of the urinary bladder pressure measurement, is not feasible for the neonatal population—especially those neonates with very low or extremely low birth weight [9]. Thus, it is necessary to establish a new non-invasive method for monitoring IAP to identify neonates at risk of ACS.

This study’s findings could potentially help change the management of patients in NICUs around the world. Continuous monitoring of IAP is essential for any neonate with risk factors for developing ACS; while monitoring IAP post reduction surgery has been shown to decrease morbidity and mortality rates of affected neonates, observations have also indicated that a timely NEC detection, through increases in IAP in the early stages of hyperinflammation, could reduce NEC morbidity and mortality [14]. One reason for the high mortality rate of NEC patients is the limited possibilities for preventing NEC progression. This is because NEC is a poorly defined condition, that, in turn, results in an inability to accurately identify subsets of premature infants at risk of developing NEC [15]. Furthermore, early detection of NEC is extremely difficult, such that it is often diagnosed too late, when necrosis and perforation of the intestines has already occurred. This limits treatment options to surgical radical bowel resection only, which is associated with a mortality rate of up to 50% [16]. 

The current gold standard for diagnosing NEC is based on criteria first established by Bell et al. in 1978. These are limited to clinical signs and symptoms and radiographic images [17]. However, in recent years, complex clinical staging systems, laboratory findings, and radiological modalities (i.e., Doppler ultrasonography) have aided in the earlier diagnosis and treatment of NEC [10]. Despite advancements in diagnostics, the disease tends to only be diagnosed once the neonate has developed actual signs and/or symptoms of NEC. Thus, identifying prospective markers specific to NEC would offer an opportunity for an earlier intervention and, consequently, improved survival. IAP monitoring should be strongly considered in all patients at risk of developing NEC. Unfortunately, the incidence of NEC increases with decreasing birth weight, making the usage of a Foley catheter for IAP monitoring purposes admissible. Thus, due to the fulminant nature of NEC, early detection, using a GC in neonates at risk, could offer opportunities for timely interventions and treatment, thus, finally, improving NEC prognosis [10]. 

In conclusion, morbidity and mortality rates could potentially be reduced through the introduction of IAP assessment using the GC, particularly in neonates, after reduction surgery of abdominal wall defects or the surgical treatment of congenital diaphragmatic hernias, as well as in primarily non-surgical conditions, like volvulus and NEC. 

However, our findings are not without limitations. Firstly, the study employs an animal testing model. As is the nature of animal testing models, measurements obtained from the abdominal wall of a fully developed rat may not accurately reflect the measurements one would obtain from human neonates [18]. Moreover, the study examined IAP and GC measurements using a sedated animal. Although patients admitted to NICUs are often premature and not very active, possible interference with the GC reading due to small movements, as well as the handling of the patient by hospital staff and caregivers, might compromise readings obtained using the GC. Secondly, it is known that the abdominal muscles of neonates, especially premature neonates, are not fully developed. Consequently, GC readings obtained from neonates of different ages will inevitably yield different measurements due to rapid changes in human development. However, as the GC solely measures changes in surface, IAP changes should still be accurately detected in this population. In fact, it is very likely that GC readings from human premature neonates might correlate more accurately with the actual IAP, as the lack of abdominal muscle activation in response to IAP increases might not mask GC readings [19]. Thirdly, as the GC does not deliver absolute values, it will be necessary to collect GC measurement reference values to reflect different age groups and time of day, and to gauge physiological fluctuation caused by, i.e., positioning of the neonate, position of the capsule, nursing, and handling by staff and caregivers (i.e., big data) [20]. Fourthly, neonates admitted to NICUs for observation and/or treatment often require medication for pain management or sedation purposes, which can relax breathing and abdominal tension, thus potentially skewing GC measurements [21,22]. 

Overall, our study suggests that the usage of the GC to detect changes in IAP is a cost-effective, non-invasive, and reliable method for monitoring IAP changes using a rat testing model. Future studies are necessary to assess the GC’s usage for IAP monitoring in the human neonatal population. If these results prove fruitful, earlier diagnosis and treatment of pathologies associated with increases in IAP could be possible, resulting in improved survival rates of sick neonates. 

## Figures and Tables

**Figure 1 children-10-01422-f001:**
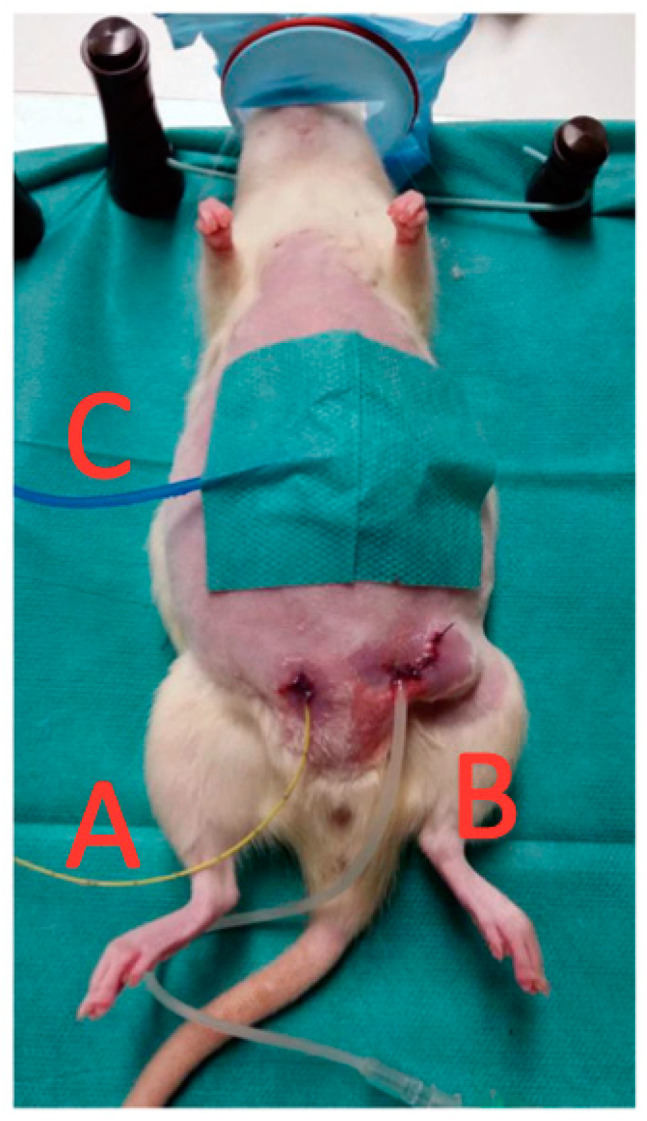
Animal experimental design set-up: After anesthesia, the subject was positioned in dorsal decubitus position and secured using rubber bands. Via two mini-laparotomies, a Codman intracranial pressure probe was placed in the right lower quadrant (**A**) and a Heidelberger extension in the left lower quadrant (**B**). The Graseby capsule was attached to the abdomen using surgical tape (**C**).

**Figure 2 children-10-01422-f002:**
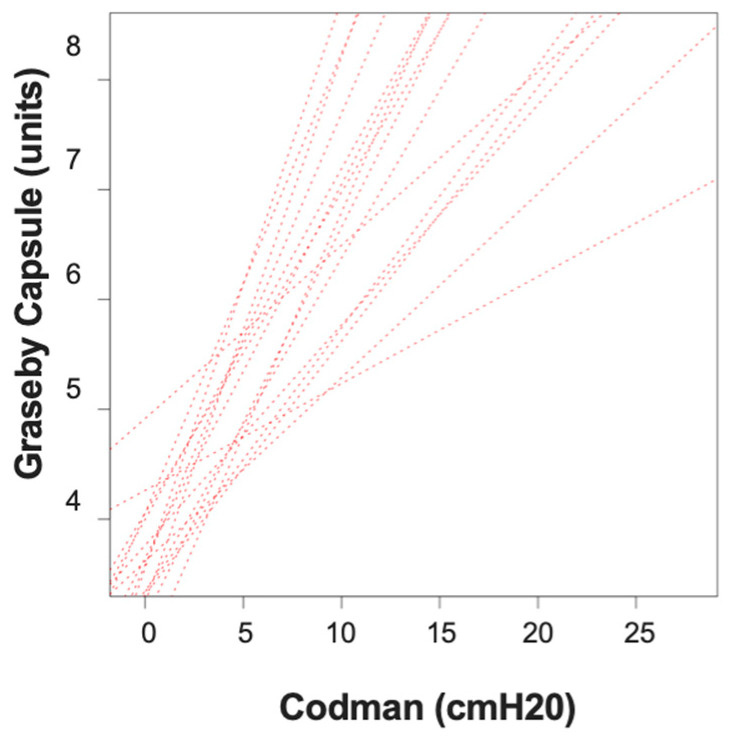
Correlation between intra-abdominal pressure changes and epidermal abdominal Graseby capsule measurements in all subjects over time.

## Data Availability

The raw data supporting the conclusions of this article will be made available by the authors without reservation.

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
