# Peer review of "Abdominal Wall Movements Predict Intra-Abdominal Pressure Changes in Rats: A Novel Non-Invasive Intra-Abdominal Pressure Detection Method"

_children, 2023, doi:10.3390/children10081422_

Round 1

Reviewer 1 Report

Thank you for reviewing this manuscript. The authors investigated in animal models a non-invasive method to measure intraabdominal pressure. In critically ill patients is crucial to measure intraabdominal pressure. The gold standard method in adults is the Foley catheter placement into the bladder. In neonates, this can be harmful. Therefore, the authors tested the Graseby capsule widely used in neonates. They found that the Graseby capsule attached to the abdomen could predict the change in the intraabdominal pressure. This non-invasive method in neonates would be really helpful. Due to the nature of the study, it was an animal experiment, and its limitation was that it was not tested on humans.

Reviewer 2 Report

The aim to looking for an alternative way to monitor intraabdominal pressure is very useful, it is an interesting work, however missing some points in the description (the results)

Change line 76-82: Following materials were used in the experiment:

·         Graseby capsule (GC)

·         Codman intracranial pressure (ICP) probe

·         Heidelberger extension, syringe, and clamp

·         Tissue adhesive (LIQUIBAND Standard, Advanced Medical Solutions Plymouth,  Ltd.) and suturing material

·         Anesthesia and surgical equipment

Line 88: Change: “nosecone”

Line 88; use “rat” for subject

Line 95: use “prior” for Prior

It is necessary to write the results not only show them in the table
